# Analysis of cholesterol in mouse brain by HPLC with UV detection

**María A. Paulazo, Alejandro O. Sodero**[ID]*

Institute of Biomedical Research (BIOMED), Pontifical Catholic University of Argentina (UCA) and National Scientific and Technical Research Council (CONICET), C1107AFF Buenos Aires, Argentina

* alejsodero@gmail.com

## Abstract

We describe a sensitive high performance liquid chromatography (HPLC)-based method for the determination of cholesterol in brain tissue. The method does not require the derivatization of the analyte and uses separation and quantification by reversed-phase HPLC coupled to UV detection. Lipids were methanol/chloroform extracted following the method of Bligh and Dyer, and separated using isopropanol/acetonitrile/water (60/30/10, v/v/v) as mobile phase. We observed lineal detection in a wide range of concentrations, from 62.5 to 2000 ng/μL, and were able to detect a significant increase in the brain cholesterol levels between postnatal days 2 and 10 in C57BL6 mice. Based on our validation parameters, we consider this analytical method a useful tool to assess free cholesterol in rodent brain samples and cell cultures.

**Data Availability Statement:** All relevant data are within the manuscript.

**Funding:** This work was supported by the Pontifical Catholic University of Argentina. The funder had no role in study design, data collection

## Introduction

Brain cholesterol represents 23% of the total body cholesterol and is relevant for brain structure and physiology. Optimal amounts of this sterol are essential to sustain important cellular functions, like the permeability of the plasma membrane, the intramembrane diffusion of receptors, the secretion of neurotransmitters, and the synthesis of steroids and myelin. Myelin is a specialized membrane with a high percentage of lipids ($\sim$70%) compared to any other cell membrane [1]. Cholesterol is particularly abundant in the myelin, representing approximately 30% of the total lipids. The high proportion of cholesterol and other lipids contribute to the efficient and rapid propagation of the action potentials in neurons. The production of myelin by the glial cells is absolutely dependent on the cholesterol biosynthesis, which involves more than 20 enzymes that convert acetate into cholesterol [2, 3, 4, 5].

From the metabolic point of view, brain cholesterol is compartmentalized in two main pools: the stable ($\sim$70%), found in the myelin membranes of the white matter, and the active ($\sim$30%), present in the plasma and subcellular membranes of neurons and glial cells of the gray matter. The concentration of cholesterol in the stable pool is high ($\sim$40 mg/g tissue) and reflects the dense packing of multiple opposed lipid bilayers in the myelin sheath, while the concentration in the active pool is lower ($\sim$8 mg/g tissue) [6].

Essentially all the brain cholesterol derives from the de novo synthesis, because this sterol cannot cross the blood-brain barrier. In adult mammals, brain cholesterol is maintained

and analysis, decision to publish, or preparation of the manuscript.

**Competing interests:** The authors have declared that no competing interests exist.

almost constant because the rates of synthesis and catabolism are similar [6]. Cholesterol is removed from the brain by catabolism to 24S-hydroxycholesterol, that crosses the blood brain barrier and is transported via the circulation to the liver for further metabolism [7].

A relevant observation is that rodents, different to humans, myelinate essentially after birth [8]. As a consequence, in the first two weeks of life, there is a significant increase in the mRNA levels of the catabolic enzyme cholesterol 24-hydroxylase (CYP46A1), that is paralleled by higher amounts of 24S-hydroxycholesterol in brain [9]. In addition, in the developing fetus, after the closure of the blood-brain barrier, and in the very young animal, desmosterol levels are high, indicating a rapid de novo synthesis of cholesterol [10].

Here, we developed a novel and reliable method to quantify cholesterol by HPLC with UV detection, after lipid extraction with methanol/chloroform [11]. Because it has been demonstrated that cholesterol rapidly increases in the mouse brain over the first days of postnatal life due to the myelination process [8], we control our quantification method measuring the cholesterol levels in the mouse brain at postnatal days 2 (P2) and 10 (P10). In agreement with previous reports [12, 13], we observed a significant increase of brain cholesterol in the first days of postnatal life.

# Materials and methods

## 2.1. Ethics statement

C57BL6 newborn mice were obtained from females grown in our animal facility. The sacrifice of the pups was done following the Guide for the Care and Use of Laboratory Animals (NRC, USA) and local regulations of the ANMAT (Argentina). The experimental protocol was approved by the Comité Institucional de Cuidado y Uso de Animales de Laboratorio (CICUAL-BIOMED 005–2019).

## 2.2. Chemicals

Isopropanol and acetonitrile, used for the preparation of the mobile phase, were HPLC grade (J.T. Baker). Ultrapure water was obtained using a Milli-Q filter system. Methanol and chloroform, utilized in the purification of lipids, were analytical grade. Cholesterol was from Sigma.

## 2.3. Samples and lipid extraction

Mouse brains were removed and placed in ice-cold HBSS without $Ca^{2+}/Mg^{2+}$. The whole tissue was dissociated in 800 μL of acid saline solution (ASS, 0.9% NaCl in 15 mM HCl) using the pipette tip. After the dissociation, 2 mL of methanol/1 mL of chloroform were added and the sample was vortexed for 20 seconds. Then, 1 mL of chloroform/1 mL of ASS were added, and the sample was vortexed, incubated for 10 minutes in ice and centrifuged at 200g for 5 minutes at 4˚C. The upper phase was discharged and the lower phase was mixed with 2 mL of methanol/1.8 mL of ASS. The sample was vortexed, incubated in ice for 10 minutes and centrifuged at 200g for 5 minutes at 4˚C. Finally, the lower chloroform phase was dried out at 40˚C using a speed-vac concentrator (Savant SPD121P, Thermo Scientific).

## 2.4. Instrumentation

The measurements were performed using a Dionex UltiMate 3000 System (Thermo Scientific) equipped with autosampler and UV detector. The chloroform phase resulting from the methanol/chloroform extraction was vacuum dried, resuspended in mobile phase and filtered through a 45 μm PDVF filter. The autosampler was set to take 50 μL of this solution. The chromatographic separation was achieved using a Phenomenex Luna C18(2) column (150×4.6

mm) packed with 5 μm silica particles. Instrument control, data acquisition and analysis were achieved with the Chromeleon software (v7.2.6, Thermo Scientific).

### 2.5. Chromatographic separation

The chromatographic separation was performed using a mobile phase containing isopropanol/acetonitrile/water (60/30/10, v/v/v), at a flow rate of 1 mL/min. The column temperature was set at 28˚C and the monitoring wavelength at 205 nm. The identification of cholesterol was done considering the full UV spectra and the retention time (RT). Cholesterol showed a RT of 8.4 minutes. To quantify the amount of cholesterol the area under the curve (AUC) was used.

### 2.6. Calibration curve

The highest standard (2 μg/μL) was prepared by solving 2 mg of cholesterol in 1 mL of ethanol. This solution was then serially diluted in mobile phase to generate the different points of the standard curve.

## Results

### 3.1. Validation of the method

The complete validation of the method was done following widely accepted analytical criteria [14, 15, 16]. The standard addition strategy was used to validate the cholesterol identification. The cholesterol peaks were identified by comparison of their retention times with those of the standards (Figs 1 and 2). The evidence that the shape and position of the cholesterol peaks were not affected by the brain matrix suggested the absence of any apparent matrix effect. This was confirmed by comparing the slopes of calibration curves made by spiking different amounts of cholesterol either in mobile phase or matrix, which were 7.82 ± 1.06 and 8.06 ± 1.59, respectively. These slopes did not differ significantly (P = 0.80; Student's t-test), confirming the absence of any matrix effect.

   The limit of detection (LOD) was determined as the concentration that produced a detector signal that could be clearly distinguished from the baseline (2 times larger than the baseline), being in our case 15 ng/μL. The limit of quantification (LOQ) was determined injecting

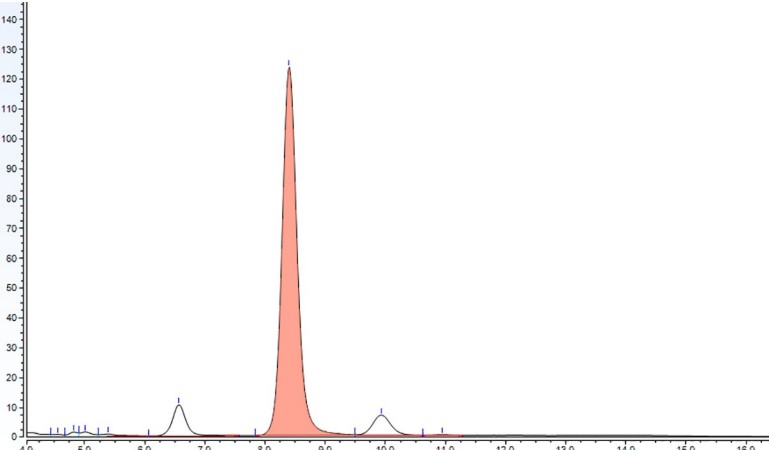

**Fig 1. Chromatogram of the cholesterol standard at 62.5 ng/μL (LOQ).** The cholesterol peak (coral colored) shows a characteristic RT of 8.4 minutes.

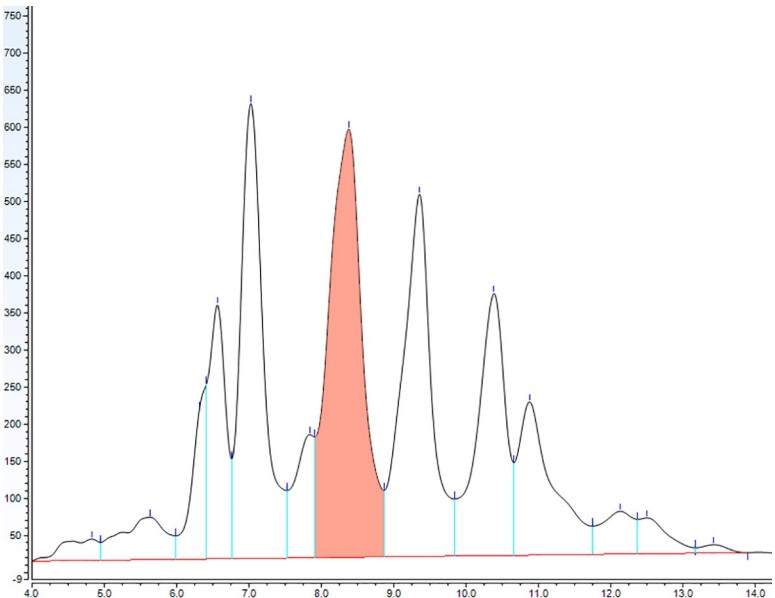

**Fig 2. Representative chromatogram showing the cholesterol detection in brain tissue (coral peak).** The dried lipids obtained after the methanol/chloroform purification were solved in 1 mL of mobile phase and 50 μL of this solution were injected for analysis.

decreasing concentrations of the analyte until the signal to noise ratio was 9:1. For our method the LOQ was 62.5 ng/μL.

The linearity of the method was tested using a 6-point calibration curve. At each concentration at least 3 replicates were assayed. The validation parameters of all fitted calibration curves were satisfactory. The average calibration curve used to determine the amount of brain cholesterol showed a $R^2$ = 0.9981 (Fig 3).

Intra- and inter-day repeatability were estimated in triplicates with cholesterol standard solutions of 62.5, 125, 250, 500 and 1000 ng/μL. All the relative standard deviations (RSD) were below 5% (Table 1), indicating that the quantification method has very good precision.

Accuracy was determined by spiking brain samples with known amounts of cholesterol, and expressed as the percentage of analyte recovery, calculated as follows:

$$Recovery\ (\%) = [(measured\ amount - endogenous\ content)/(added\ amount)] \times 100$$

Our method showed a satisfactory analyte recovery of 86 ± 11% (n = 3).

In addition, to evaluate the robustness of the method we analyzed two variables that may impact its performance: the monitoring wavelength and the column temperature. The cholesterol RT stayed constant when we simultaneously monitored the signal at two different wavelengths, 205 and 208 nm. We decided to set 205 nm as the optimal wavelength because the obtained AUC was slightly higher than the AUC at 208 nm. Setting the column temperature either at 22˚C or 28˚C did not have any impact on the RT or AUC.

## 3.2. Application of the method to brain samples

After the analytical validation of the method, we measured the cholesterol levels in brains of C57BL6 mice at postnatal days 2 (P2) and 10 (P10). In this interval of the postnatal life, the amount of cholesterol increased from 3.7 ± 0.4 to 6.2 ± 0.7 μg/mg of wet tissue (p < 0.05, Student's t-test; Fig 4). This 68% increase is in agreement with previous findings showing a higher cholesterol biosynthesis after birth in mouse and rats [6].

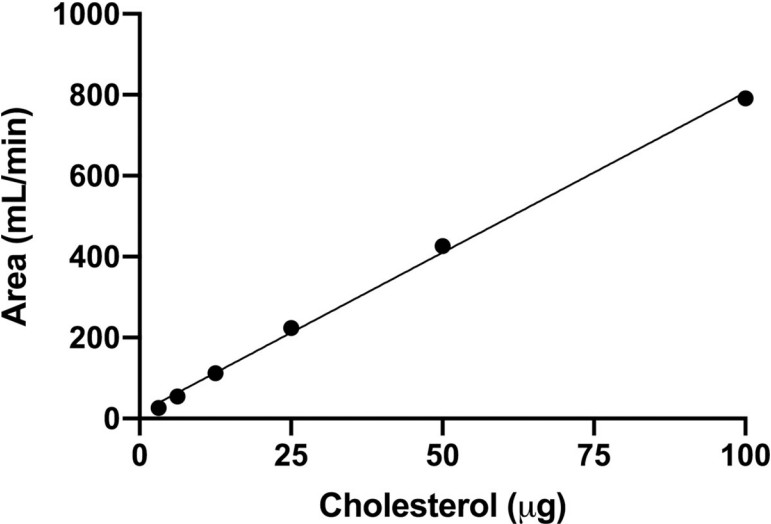

**Fig 3. Calibration curve used to determine the amount of cholesterol in the experimental samples.** The linear regression showed very good correlation ($y = 7.89\ x + 13.47$; $R^2 = 0.9981$).

## Discussion

In mice, the levels of brain cholesterol progressively increase from birth to 3 months of age, reaching approximately 150 micrograms per milligram of wet tissue (see Table 2). In the present work, using HPLC with UV detection, we report similar cholesterol amounts in the early postnatal mouse brain (0–10 micrograms per milligram of wet tissue). In addition, we found a significant increase in the levels of brain cholesterol in 10 days-old mice, compared to 2 days-old mice, most likely due to the higher cholesterol biosynthesis required for myelin formation in this early postnatal period.

The detection of brain cholesterol changes has relevance to understand the neurobiology of several disorders associated to dysfunctions in the homeostasis of this lipid [17, 18, 19, 20, 21]. Traditionally, cholesterol has been quantified in different brain preparations by the cholesterol oxidase method, coupled to colorimetric or fluorometric detection, LC/MS or GC/MS (see Table 3). Here, we describe a different way to analyze free cholesterol in brain samples from which lipids are extracted using methanol/chloroform [11]. Compared with a previously reported method [22], we found out higher signals monitoring the absorbance at 205 nm instead of 200 nm. Another difference is that we do not perform an Abell-Kendall saponification of the sample, followed by lipid extraction with petroleum ether or hexane. Instead, we use methanol and chloroform for lipid extraction, which can be manipulated and disposed in an easier way than petroleum ether or hexane. In addition, we were able to detect low amounts

**Table 1. Intra- and inter-day precision of the cholesterol determination.** The relative standard deviation (RSD) at five concentration levels of the cholesterol standard is shown.

| Concentration level (ng/μL) | Intra-day RSD (n = 3) | Inter-day RSD (n = 3) |
|---|---|---|
| 62.5 | 2.3% | 3.6% |
| 125 | 1.4% | 2.3% |
| 250 | 0.4% | 2.1% |
| 500 | 4.9% | 2.5% |
| 1000 | 4.2% | 4.1% |

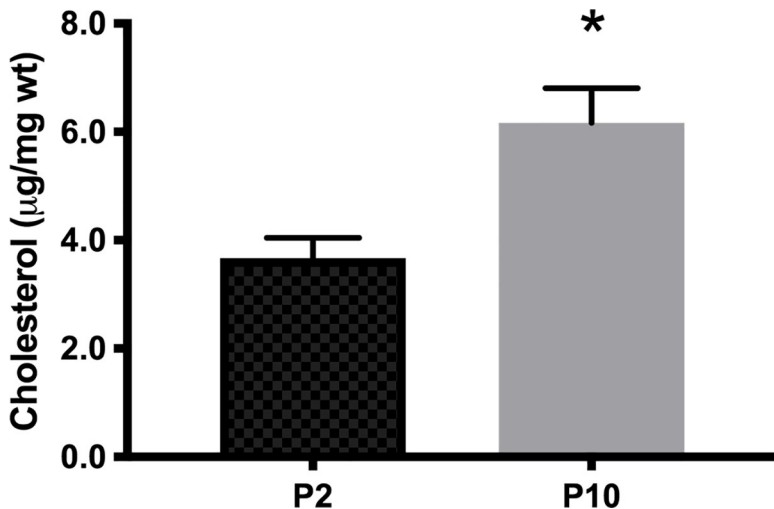

**Fig 4. Cholesterol concentrations in whole brains of C57BL6 mice at postnatal days 2 (P2) and 10 (P10).** Data are expressed as µg of cholesterol/mg of wet tissue (µg/mg wt). Bars represent mean ± SEM (n = 3); * $p < 0.05$, Student's t-test.

**Table 2. Comparison of brain cholesterol levels determined by different methodologies.** The reported cholesterol concentrations in early life are similar to the amounts measured by HPLC with UV detection in this work.

| Report | Mouse age | Quantification method | Measured cholesterol (µg/mg wet tissue) |
|---|---|---|---|
| Marcos J. et al. [12] | 1 day | GC-MS | 2.5 |
| Marcos J. et al. [12] | 14 days | GC-MS | 6 |
| Meljon A. et al. [13] | 1 day | LC-MS | 2.2 |
| Meljon A. et al. [13] | 15 weeks | LC-MS | 16 |
| Lu F. et al. [24] | 9 days | Enzymatic | 8.5 |
| Nunes V.S. et al. [25] | 12 weeks | GC-MS | 120 |

**Table 3. Different analytical approaches commonly used to determine cholesterol in biological samples.** The type of detection and the limit of quantification (LOQ) of each methodology are shown.

| Method | Detection | LOQ (ng/µL) |
|---|---|---|
| Enzymatic/colorimetric [26] | 500 nm (quinoneimine) | 250 |
| Enzymatic/fluorometric [27] | 585 nm (resorufin) | 0.1 |
| HPLC-UV [present work] | 205 nm (cholesterol) | 62.5 |
| LC-MS | m/z | < 50 |
| GC-MS | m/z | < 50 |

of cholesterol in different cell lines in culture, indicating that our method is also suitable to quantify free cholesterol in small pieces of tissue, like mouse brain structures.

Although GC or LC separation associated to MS detection remain the most sensitive methodologies for cholesterol quantification in tissues and cultured cells [23], the here reported HPLC method with UV detection represents a reliable alternative to measure cholesterol in biological samples.

## Author Contributions

**Conceptualization:** María A. Paulazo, Alejandro O. Sodero.

**Investigation:** María A. Paulazo, Alejandro O. Sodero.

**Supervision:** Alejandro O. Sodero.

**Writing – original draft:** María A. Paulazo.

**Writing – review & editing:** Alejandro O. Sodero.

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
