## [Decision Letter · Decision Letter 0]

4 Nov 2019

PONE-D-19-26047

Analysis of cholesterol in mouse brain by HPLC with UV detection

PLOS ONE

Dear Alejandro O. Sodero,

Thank you for submitting your manuscript to PLOS ONE. After careful consideration, we feel that it has merit but does not fully meet PLOS ONE’s publication criteria as it currently stands. Therefore, we invite you to submit a revised version of the manuscript that addresses the points raised during the review process.

We would appreciate receiving your revised manuscript by 13th December. To enhance the reproducibility of your results, we recommend that if applicable you deposit your laboratory protocols in protocols.io, where a protocol can be assigned its own identifier (DOI) such that it can be cited independently in the future. For instructions see: http://journals.plos.org/plosone/s/submission-guidelines#loc-laboratory-protocols

We look forward to receiving your revised manuscript.

Kind regards,

Tommaso Lomonaco, Ph.D

Academic Editor

PLOS ONE

Journal Requirements:

2. To comply with PLOS ONE submissions requirements, please provide methods of sacrifice in the Methods section of your manuscript.

Additional Editor Comments:

Dear Authors,

the submitted paper requires major revisions in order to address all the questions and suggestions proposed by the reviewers. In particular all the analytical figures of merit of the developed method must be included in the revised version of the paper.

Please, include specific quality parameters as recovery, matrix effect, inter- and intra-day recovery and precision, LOD and LOQ and stability. I suggest to include a table in order to highlight the main differences with other analytical approached already published in the literature. The authors should discuss in detail the main advantages of the proposed method. Moreover, I suggest to consider the following papers as example for the method validation and include them in the references.

DOI: 10.1371/journal.pone.0028182

DOI: 10.1016/j.chroma.2013.08.091

DOI: 10.1371/journal.pone.0114430

Best regards,

Tommaso Lomonaco

Reviewers' comments:

Reviewer's Responses to Questions

**Comments to the Author**

1. Is the manuscript technically sound, and do the data support the conclusions?

Reviewer #1: Yes

Reviewer #2: No

2. Has the statistical analysis been performed appropriately and rigorously? 

Reviewer #1: Yes

Reviewer #2: No

3. Have the authors made all data underlying the findings in their manuscript fully available?

Reviewer #1: Yes

Reviewer #2: No

4. Is the manuscript presented in an intelligible fashion and written in standard English?

Reviewer #1: Yes

Reviewer #2: Yes

5. Review Comments to the Author

Reviewer #1: In this manuscript, the authors María Alejandra Paulazo & Alejandro Omar Sodero proposed a RP-HPLC-UV method for the analysis of cholesterol in mouse brain. The experimental design and the result are reasonable and reliable.

However, there are some points which need to be further considered.

Major comments:

1. At the end of the introduction (P4), authors need to give the other literature reports of the cholesterol determination by instrumental techniques.

2. As the authors developed and validated a RP-HPLC method for cholesterol, it is suggested to perform and include the other validation parameters like system suitability, specificity, ruggedness, robustness, etc.

3. Provide the data table(s) which includes the chromatographic and validation parameters.

4. Whether the authors cross-checked the peak purity of the cholesterol peak at RT 8.4 in Figure 2? Because as the sample run contains many peaks due to impurities in it, there are all the possibilities that the other impurities peaks may overlapped on it. It can be overruled by examining the peak purity of cholesterol (with the help of chromatographic software).

Minor comments:

1. P2, line 20; Abstract: HPLC to be expanded instead of in line 22.

5. P4, line 64-65, authors are claiming that the proposed method is in agreement with previous reports; they need to provide the literature for the previous reports.

6. P4, line 75, make +2 state of Ca/Mg superscript.

7. P5, line 96, authors claiming temperature was set at 28oC. It is not clear about the temperature of which was set at 28oC. Be clear about it. I think it is the column temperature.

8. P7, line 128, make 2 of R2 superscript.

9. P8, line 158, the sentence after the reference [18], is not correct.

10. P11, line 220, make 2 of R2 superscript.

11. Mention the name and RT on the cholesterol peak in the Figures 1 & 2.

Reviewer #2: HPLC analysis of cholesterol in biological samples had long been reported and the manuscript described a traditional method with limited analytical performance. Additionally, the validation of the method was insufficient. The presented data did not support the claim of "very good precision and accuracy". A recovery of 74% might be too low for cholesterol analysis. The specificity of the method was unknown.

6. PLOS authors have the option to publish the peer review history of their article (what does this mean?). If published, this will include your full peer review and any attached files.

Reviewer #1: Yes: Manjunatha D H

Reviewer #2: No

---

## [Author Response · Author response to Decision Letter 0]

12 Dec 2019

Dear Editor and reviewers,

We very much appreciate the criticisms and suggestions, which have led to this revised version of our manuscript entitled “Analysis of cholesterol in mouse brain by HPLC with UV detection”.

We have answered point-by-point all the major and minor comments in the document "Response to the reviewers".

Hoping that in the actual conditions our manuscript is suitable for publication,

Best regards,

Dr. Alejandro O. Sodero

---

## [Decision Letter · Decision Letter 1]

31 Dec 2019

PONE-D-19-26047R1

Analysis of cholesterol in mouse brain by HPLC with UV detection

PLOS ONE

Dear Dr Alejandro O. Sodero,

Thank you for submitting your manuscript to PLOS ONE. After careful consideration, we feel that it has merit but does not fully meet PLOS ONE’s publication criteria as it currently stands. Therefore, we invite you to submit a revised version of the manuscript that addresses the points raised during the review process.

We would appreciate receiving your revised manuscript by 5th January 2020. To enhance the reproducibility of your results, we recommend that if applicable you deposit your laboratory protocols in protocols.io, where a protocol can be assigned its own identifier (DOI) such that it can be cited independently in the future. For instructions see: http://journals.plos.org/plosone/s/submission-guidelines#loc-laboratory-protocols

We look forward to receiving your revised manuscript.

Kind regards,

Tommaso Lomonaco, Ph.D

Academic Editor

PLOS ONE

Additional Editor Comments (if provided):

Dear Authors, the paper requires minor revision before to be accepted in PlosOne journal. The analytical figure of merits included in the new version of the paper improved the quality of the manuscript. I suggest to check all the digit number in the paper, which should be approximate according to the variability of the analytical method. Moreover, the authors should include a table in the discussion in order to compare different analytical approaches commonly used to determine cholesterol in biological samples. In addition, the following articles should be included in the references with the aim to help other readers to compare the process for method validation.

DOI: 10.1371/journal.pone.0028182

DOI: 10.1016/j.chroma.2013.08.091

DOI: 10.1371/journal.pone.0114430

Best regards,

Tommaso Lomonaco

Reviewers' comments:

Reviewer's Responses to Questions

**Comments to the Author**

1. If the authors have adequately addressed your comments raised in a previous round of review and you feel that this manuscript is now acceptable for publication, you may indicate that here to bypass the “Comments to the Author” section, enter your conflict of interest statement in the “Confidential to Editor” section, and submit your "Accept" recommendation.

Reviewer #1: All comments have been addressed

2. Is the manuscript technically sound, and do the data support the conclusions?

Reviewer #1: Yes

3. Has the statistical analysis been performed appropriately and rigorously? 

Reviewer #1: Yes

4. Have the authors made all data underlying the findings in their manuscript fully available?

Reviewer #1: Yes

5. Is the manuscript presented in an intelligible fashion and written in standard English?

Reviewer #1: Yes

6. Review Comments to the Author

Reviewer #1: (No Response)

7. PLOS authors have the option to publish the peer review history of their article (what does this mean?). If published, this will include your full peer review and any attached files.

Reviewer #1: Yes: Manjunatha D H

---

## [Author Response · Author response to Decision Letter 1]

7 Jan 2020

Dear Editor,

It is a pleasure to submit for your consideration the revised version of our manuscript “Analysis of cholesterol in mouse brain by HPLC with UV detection”.

We have addressed your minor comments and modified the previous version accordingly.

Hoping the current version fulfills the criteria for publication.

Best regards,

Dr. Alejandro O. Sodero

---

## [Editor Report · Decision Letter 2]

9 Jan 2020

Analysis of cholesterol in mouse brain by HPLC with UV detection

PONE-D-19-26047R2

Dear Dr. Alejandro O. Sodero,

We are pleased to inform you that your manuscript has been judged scientifically suitable for publication and will be formally accepted for publication once it complies with all outstanding technical requirements.

With kind regards,

Tommaso Lomonaco, Ph.D

Academic Editor

PLOS ONE

Additional Editor Comments (optional):

Dear Authors, the paper is available to be published in PloSone.

Best regards.

Tommaso Lomonaco
---

## [Editor Report · Acceptance letter]

15 Jan 2020

PONE-D-19-26047R2 

Analysis of cholesterol in mouse brain by HPLC with UV detection 

Dear Dr. Sodero:

I am pleased to inform you that your manuscript has been deemed suitable for publication in PLOS ONE. Congratulations! Your manuscript is now with our production department. 

With kind regards,

on behalf of

Dr. Tommaso Lomonaco 

Academic Editor

PLOS ONE